# Temperature Areas of Local Inelasticity in Polyoxymethylene

**DOI:** 10.3390/polym16243582

**Published:** 2024-12-21

**Authors:** Viktor A. Lomovskoy, Svetlana A. Shatokhina, Raisa A. Alekhina, Nadezhda Yu. Lomovskaya

**Affiliations:** Frumkin Institute of Physical Chemistry and Electrochemistry, Russian Academy of Sciences (IPCE RAS), Leninskiy Prospekt 31, 119071 Moscow, Russia; lomovskoy49@gmail.com (V.A.L.); rioraya9@gmail.com (R.A.A.); lomo335@yandex.ru (N.Y.L.)

**Keywords:** polyoxymethylene, internal friction spectra, temperature–frequency dependencies, local dissipative processes, relaxation time, shear modulus defect, dissipation mechanisms

## Abstract

The spectra of internal friction and temperature dependencies of the frequency of a free-damped oscillation process excited in the specimens of an amorphous–crystalline copolymer of polyoxymethylene with the co-monomer trioxane (POM-C) with a degree of crystallinity ~60% in the temperature range from −150 °C to +170 °C has been studied. It has been established that the spectra of internal friction show five local dissipative processes of varying intensity, manifested in different temperature ranges of the spectrum. An anomalous decrease in the frequency of the oscillatory process was detected in the temperature ranges where the most intense dissipative losses appear on the spectrum of internal friction. Based on phenomenological model representations of a standard linear solid, the physical–mechanical (shear modulus defect, temperature position of local regions of inelasticity) and physical–chemical (activation energy, discrete relaxation time, intensities of detected dissipative processes) characteristics of each local dissipative process were calculated. It was found that the intensities of dissipative processes remain virtually unchanged for both annealed and non-annealed samples. The maximum variation in the shear modulus defect is 0.06%. Additionally, according to computational data, small changes are also characteristic of the following parameters: the activation energy varies from 0.5 to 1.4 kJ/mol and the relaxation time changes from 0.002 to 0.007 s, depending on the presence or absence of annealing. As a result of annealing, there is a significant increase in the relaxation microinheterogenity of the polymer system across the entire temperature range (250% for the low-temperature region and 115% for the high-temperature region).

## 1. Introduction

Polyoxymethylene (POM) is one of the most important engineering polymers [1,2,3] used in various industries along with polyethylene (PE). Due to the fact that POM has excellent comprehensive properties (e.g., good mechanical properties [4,5], lubrication performance [1,6], fatigue resistance [1], and dimensional stability [7]) and especially excellent process characteristics [8,9], it has found more and more practical applications in recent years [10,11,12].

Scientifically, these polymer systems differ in that POM is a heterochain polymer and PE is a homochain polymer.
-[CH2-CH2]n-PE-[CH2-O]n-POM-H…-[CH2-O]p-[CH2-CH2-O-]q-…POM-C [13]

Early study of the spectra of the internal friction in polyethylene allowed us to reveal four local dissipative processes of varying intensities that manifested in different temperature ranges of these spectra [14,15,16] (Figure 1). The most complex and the most intensive dissipative process is the $\beta_{k}$ process. This process is determined by the mobility of structural elements of the amorphous phase adjacent to the boundaries of various crystalline formations in the amorphous–crystalline PE structure, as well as by the vibrational mobility of chain sections in the crystalline phase. It can be decomposed into several dissipative processes superimposed on each other by temperature, which confirms the data of studies obtained by other methods [17]. The second intensive peak of dissipative losses β located on the spectrum $\lambda =f\left(T\right)$ in the temperature range from −140 °C to −90 °C is also, like the $\beta_{k}$ process, a complex process. The structural mechanism of this dissipative process is determined by the mobility of the chain link in the amorphous phase and, like the $\beta_{k}$ process, is characterized by the relaxation mechanism of internal friction. The process associated with a glass transition of the amorphous PE phase (α-process) is not clearly visible on the spectrum $\lambda =f\left(T\right)$ due to its total absorption by the low-temperature area of the process, which characterizes the difference of conformational positions of passing chains in amorphous intercrystalline PE layers, which, in turn, leads to the formation of several structural–kinetic subsystems built of similar structural units [14,15,16,18,19,20,21,22]. In addition, the spectrum $\lambda =f\left(T\right)$ of the freely damped oscillatory process (Figure 1) reveals a region with a very weak intensity of local dissipative processes μ, a process observed in the ~−70 °C temperature region.

For POM (both homopolymer and copolymer obtained in various technological processes), such studies have not been carried out. It should be noted that POM-H (homopolyoxymethylene) and POM-C (polyoxymethylene copolymer) have different structures due to the presence of a co-monomer, trioxane in POM-C, which leads to the complication of the structure of POM-C relative to POM-H [13,23,24,25] and PE and differences in their physical and mechanical characteristics (Table 1) [26,27,28].

Polyoxymethylene has a high degree of crystallinity χ≈60÷80 % (for a copolymer); however, due to the fact that the POM structure also contains an amorphous phase, the voids may be formed due to different densities and thermal expansion coefficients between the amorphous and crystalline phases [31,32].

The crystalline phase can be represented by models that characterize the structure of this phase as follows:Extended-chain crystals (ECCs) 
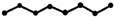
;Folded-chain crystals (FCCs) 
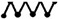
—lamellar crystals;A hybrid structure of the ECC-FCC sequence, i.e., shish-kebab structure [33,34,35,36].

The melting point T_m_(FCC) of the folded-chain crystals is lower than the melting point of the extended-chain crystals T_m_(ECC).
Т_m(FСС)_ < Т_m(ЕСС)_.

The lamellar crystals (FSSs) are characterized by the lamellar thickness L-length of the macromolecule chain section between successive lamellar folds occurring along the chain [33,34,35,36].

The crystalline phase in the POM structure is defined by two modifications differing in lattice structure as follows:The orthorhombic structure is characterized by a helical conformation of “two monomer units per helical turn of the macromolecule chain”, i.e., 2/1. In the orthorhombic structure, the unit cell is formed by two chains of macromolecules, each of which is included in the formation of the unit cell by two monomer units [37,38]. The orthorhombic modification is not common and easily transforms into the hexagonal modification when heated [39].When POM crystallizes from the melt, a spherulitic morphology appears in the structure [40,41].

Previous studies of relaxation phenomena in POM revealed several dissipative processes in the temperature range of −160 °C ÷ + 180 °C [42,43,44,45].

The ***β* process** is the mobility of macromolecule chain links in the crystalline regions of POM, which is split by temperature into a number of small-scale dissipative processes in the range from −100 °C to −50 °C. This process is observed in the temperature range T_g_ < T < T_m_.

The mobility of chain links is observed between lamellar formations at T ≈ −10 °C for the amorphous phase C [44].

The ***α*-relaxation process** is the mobility of segments and a change in the conformation of macromolecules, which is allowed as a large-scale movement of significant sections of the chain in the crystalline phase of POM and segmental mobility in the amorphous phase of POM. The α-relaxation in the crystalline phase is observed at T ≈ +120 °C. At +20 °C ≤ T ≤ +60 °C, apparently, there is a process associated with the mobility of boundaries between small mosaic crystalline blocks and at +60 °C ≤ T ≤ +100 °C—a process associated with the mobility of the mosaic crystal nucleus.

The ***γ*-relaxation process** is observed in the region T ≈ −70 °C and is associated with local reorientation in the form of twisting of the main chains in defective crystalline formations. The assumption is that this process can be divided into several superimposed dissipative processes, where at T ≈ −90 °C, there can be torsional oscillations of the chains of POM macromolecules in the amorphous phase [44].

Earlier studies of relaxation phenomena in POM based on internal friction spectra carried out by different authors [42,43,44,45] did not provide convincing evidence for the mechanisms of local dissipative processes at temperatures manifested on $\lambda =f\left(T\right)$ and could not give an unambiguous atomic–molecular interpretation of specific processes.

Nevertheless, these studies have shown that three loss peaks of complex shapes can be observed on the temperature dependencies $\lambda =f\left(T\right)$ due to their possible expansion into constituent peaks having a different atomic–molecular nature of their origin. In addition, one of the main sources of the obtained contradictory data is the lack of accurate data on the physical and mechanical characteristics of the studied POM specimens (nature of POM-H or POM-C, degree of crystallinity χ, molecular weight M_w_, exact T_g_ value, etc.).

In this paper, an in-depth analysis of the situation on relaxation spectrometry of polyoxymethylene in general and POM-C separately has been carried out. It was found that the processes manifested in the spectra of earlier works had a similar pattern, but the temperature position of dissipative loss peaks, their labeling, intensities, and interpretation differ from each other, which indicates the relevance of the presented study. In addition, the difference in polyoxymethylene due to the presence or absence of copolymers has not been taken into account previously.

Herein, a study and theoretical analysis of local (temperature localized in a specific temperature interval) dissipative processes was carried out by an internal friction spectrum and temperature dependencies of the frequency of a freely damped vibrational process excited in amorphous–crystalline polyoxymethylene samples.

It should be noted that the study of POM-C in dynamic modes by the free-damped oscillatory process method is carried out for the first time. The temperature range of this study is wide. As a result of the improvement of the setup [46,47] with the use of computer data processing, local processes of medium intensity were obtained and investigated in detail for the first time, which was not possible before. In addition, it became possible to calculate the shear modulus defects for each local dissipative process in different temperature intervals by temperature changes in the frequency of a free-damped oscillatory process. Previously, these issues were not covered, which indicates the novelty of the studies and the results obtained.

The temperature–frequency dependencies of the free-damping process obtained simultaneously with the internal friction spectra make it possible to determine the temperature position and the magnitude of local dissipative processes in the entire temperature range, determine the nature of these processes, and draw a conclusion about the ability of the material to resist external shear effects. Thus, by comparing experimental and calculated data, it is possible to make a prediction about the behavior of polymer material in a wide range of temperatures.

To achieve this goal, the following tasks were solved:
We studied the internal friction spectra in the temperature range of −150 °C ÷ +160 °C and identified the local temperature regions of inelasticity;We obtained temperature dependencies of the frequency of the oscillatory process excited in POM specimens;We studied the influence of heat treatment (annealing) of the original sample on the dependencies $\lambda =f\left(T\right)$ and $\nu =f\left(T\right)$;We calculated the physical–mechanical (shear modulus, shear modulus defect) and physical–chemical characteristics (activation energy of dissipative processes, discrete relaxation times, determination of the mechanisms of each dissipative process (relaxation, phase, hysteresis));We determined the influence of temperature conditions of the external thermal field (annealing) on the calculated physical–mechanical and physical–chemical characteristics of each individual dissipative process and the investigated polyoxymethylene system as a whole.

## 2. Experimental

### 2.1. Materials and Heat Treatment Methods

The material used in this study was Mascon POM 27 polyoxymethylene (POM). The main characteristics of this POM grade are shown in Table 2.

The manufacturing of POM plates with a thickness of about 1 mm was carried out by pressing in a closed-type mold at T = 180 °C using a hydraulic press P-10 “Tekhmash”, Russia. Subsequently, specimens in the rectangular cross-section with a thickness of 1mm, a length of 65 mm, and a width of 5 mm were cut from these plates.

In this work, two specimens are presented and are designated POM, specimen 1(1), original and POM, and specimen 1(3), annealed. To remove the internal stresses arising in the process of obtaining POM samples, annealing was carried out for subsequent comparative analysis of the experimental results of the original and annealed specimens. Annealing was carried out at a temperature $T\approx \left(75÷90\right)\%\cdot T_{m}$, where $T_{m}$ is the melting point [49,50,51,52]. The melting point in this paper was determined from the experimental thermograms (DSC) obtained (presented in the experimental part).

Annealing was carried out in a Binder FD 115 drying chamber, where specimens were heated at a rate of 1°/min to 135° for 2.5 h; then, specimens were held for 10 min, followed by cooling for 24 h.

The structure of polyoxymethylene films was studied using a polarizing optical microscope model “MP-7” No. 70193.

Measurements using differential scanning calorimetry (DSC) were conducted with a DSC Q100 unit of Intertech Corporation (USA) at a rate of 5 °C/min at an argon current of 50 mL/min.

The melt flow index was determined using an extrusion rheometer PTR-LAB-02 (JSC LOIP, Russia).

### 2.2. Relaxation Spectrometry Method

Experimental temperature dependencies of dissipative losses (internal friction spectra $\lambda =f\left(T\right)$) and temperature dependencies of changes in the frequency $\nu =f\left(T\right)$ of a free-damped oscillation process excited in a specimen of the material under study were obtained on an installation that is a horizontal pendulum design [46,47].

As a measure of internal friction in this case, the logarithmic decrement of the damped oscillation process can be used (Figure 2), which enters the equation describing the oscillatory process in the form of a ratio as follows:(1)$$\varphi (t)=\varphi_{\max}\;\exp^{-\beta \theta }=\varphi_{\max}\;\exp\;\left[-\left(\frac{\lambda }{\pi }\right)\;t\right],$$
where $\varphi_{\max}$ and φ(t) are, respectively, the amplitude (maximum) and the current value of the twist angle of the unattached end of the test specimen; β is the damping coefficient; θ is the oscillation process period; and t is the time.

The study of many physical and mechanical characteristics in materials of different chemical natures and structures is based on the study of the parameters of transition processes from equilibrium or disequilibrium to another equilibrium. The dynamic characteristics of these transients are evaluated by the characteristics of the transition functions, which represent the response of the system both to changes within the system itself and to changes in external impacts on this system. By the intensity of changes in the parameters of the transition functions, it is possible to obtain certain information about the structure of the systems under study. The presence of transient processes from disequilibrium to equilibrium of a non-conservative system leads to thermodynamic irreversibility of this process and, as a consequence, to dissipation (internal friction) in the system under study of a part of energy of external force impact. A quantitative characteristic of energy dissipation in the system is the absorption coefficient, which is related to other dissipative characteristics by the following relation:(2)$$\Psi =\frac{\Delta W}{2\pi W}=Q^{-1}=tg\delta =\frac{\lambda }{\pi },$$
where ΔW and W represent the amounts of irreversibly dissipated energy and energy supplied to the system, respectively; $Q^{-1}$ is the internal friction; δ is the phase shift angle between the external influence and the system’s response to that influence; and λ is the logarithmic decrement of the damping oscillatory process [55].

This makes it possible to obtain experimental spectra of internal friction. In relation (2), the logarithmic decrement of the oscillatory process can be determined in the following form:(3)$$\lambda =\frac{1}{n}\ln\frac{\varphi_{\max}}{\varphi \left(t\right)},$$
where n is the number of oscillations between the oscillation having amplitude $\varphi \left(t\right)$ and the oscillation having amplitude $\varphi_{\max}$ in the temporary development of the oscillatory process (Figure 2b).

For each temperature T in this study, there will be a different temporary development of the oscillatory process (relation (3)) and a different value of the logarithmic decrement λ, as well as a different value of the frequency ν of the oscillatory process excited in the specimen under study.

## 3. Results and Discussion

The following results were obtained in the course of the work.

### 3.1. Internal Friction Spectra $\lambda =f\left(T\right)$ and Temperature Dependencies of Frequency $\nu =f\left(T\right)$

Figure 3 shows the obtained spectra $\lambda =f\left(T\right)$ and temperature dependencies of the frequency of the materials $\nu =f\left(T\right)$ under study, from which it follows that these spectra are characterized by five dissipative processes *(I–V*) that are located in different temperature ranges and have differences in the intensity of dissipative losses $\lambda_{\max}$. The most intense processes are *II* and *IV*. These peaks of dissipative losses are complex, that is, they are composed of at least two processes superimposed on each other.

The temperature dependencies of frequency $\nu =f\left(T\right)$ of the free-damped oscillatory process for the investigated POM specimens (Figure 3b) reveal a significant deviation from the linear dependence in those temperature regions where $\lambda =f\left(T\right)$ the dissipative loss peaks on the *II* and *IV* spectra. However, a more detailed examination of this local frequency change, carried out when determining shear modulus defects ΔG after the theoretical expansion of experimental complex loss peaks according to a normal Gaussian distribution, made it possible to identify anomalies for other hidden (as a result of being superposition on each other) dissipative processes.

### 3.2. POM Structure According to Polarization Optical Microscopy Data

The structure of polyoxymethylene films was studied using a polarizing optical microscope. Figure 4 shows that POM is an amorphous–crystalline copolymer consisting of an amorphous phase and ordered spherulite formations as follows:POM spherulites obtained by cooling from the melt at temperatures well below 150 °C exhibit a distinct Maltese cross;When cooling at temperatures close to or slightly above 150 °C, the structure of the crystalline phase becomes dendritic, and the Maltese cross is barely visible [38].

### 3.3. DSK Method

DSC thermograms of the original and annealed samples revealed one endothermic process (Figure 5), which made it possible to determine the degree of crystallinity χ of the samples according to ratio (4) as follows:(4)χ=ΔHmΔHm0⋅100%,
where $\Delta H_{m}$ is the enthalpy of fusion of the studied POM specimens and $\Delta H_{m}^{0}=326\;J/g$ is the theoretical value of enthalpy of fusion of POM with a 100% degree of crystallinity [56,57,58,59,60]. The obtained values of the degree of crystallinity were in the order of 51 ÷ 53%. However, in some works, it is stated $\Delta H_{m}^{0}=186\;J/g$ [61,62], and (even $\Delta H_{m}^{0}=260\pm 30\;J/g$ [63]) that calculation gives values of the degree of crystallinity in the order of 90 ÷ 94%. Such high degrees of crystallinity are uncharacteristic for a polyoxymethylene copolymer (commercial at that); accordingly, the values of the degree of crystallinity were adopted by taking into account $\Delta H_{m}^{0}=326\;J/g$. The calculated values of the enthalpy of fusion $\Delta H_{m}$ and the degree of crystallinity χ of the original and annealed POM are given in Table 3.

### 3.4. POM and the Melt Flow Index

The molecular weight of the system under study was determined using the melt flow index (MFI), which was measured using an extrusion rheometer. The calculation of average molecular weight ($M_{W}$) for the polyoxymethylene copolymer [23,28] was carried out using the following formula:(5)$$\begin{array}{l} MFI=3.3\cdot 10^{18}\cdot M_{W}^{-3.55}\\ M_{W}=\sqrt[{3.55}]{\frac{3.3\cdot 10^{18}}{27}}=65055.15\approx 6.5\cdot 10^{4} \end{array},$$
where MFI is the melt flow index measured with a deposited mass of 2.16 kg at 190 °C [64].

## 4. Theoretical Analysis of the Results Obtained by Relaxation Spectrometry

An expansion of the most intense complex dissipative loss peaks on the spectrum $\lambda =f\left(T\right)$ using the Gaussian normal distribution in Origin Pro 2017 (b9.4.0.220) software was performed, as shown in Figure 6.

As a result of this expansion, a number of processes of different intensities were identified, indicated on the spectra by Roman numerals *I*, *II*, … *V* (Figure 6 and Figure 7).

Each peak of dissipative losses in the spectrum can be described by the differential equation of the phenomenological model of a standard linear solid as follows:(6)$$\frac{d\sigma }{dt}+\frac{G_{1}}{\eta }\sigma =\left[\left(G_{1}+G_{2}\right)i\omega +\frac{G_{1}G_{2}}{\eta }\right]\gamma_{0}\exp\left(i\omega t\right),$$
where σ is the stress arising in the system under study; $G_{1}$ and $G_{2}$ are the shear modules of the subsystem causing the appearance of the peak of dissipative losses on the spectrum $\lambda =f\left(T\right)$ and the aggregate subsystem responsible for the shape and rigidity of the sample of the system under study as a whole; η is the viscosity of the structural-kinetic subsystem causing the appearance of the peak of dissipative losses on the spectrum $\lambda =f\left(T\right)$; and ω is the circular frequency of the oscillatory process excited in the sample.

The solution of the differential equation in the dynamic regime of a damping oscillatory process can be represented as follows [55,65]:(7)$$\lambda_{i}=2\lambda_{i_{\max}}\frac{\omega \tau_{i_{\max}}}{1+{\left(\omega \tau_{i_{\max}}\right)}^{2}},$$
where $\lambda_{i}$ and $\lambda_{i_{\;\max}}$ are the current and maximum values of the logarithmic coefficient of the damping oscillatory process for the dissipative process and $\tau \equiv \tau_{i}=\frac{\eta }{G_{1}}$ is the relaxation time of the subsystem causing the appearance of the dissipative loss peak on the spectrum $\lambda =f\left(T\right)$. It follows from relation (7) that the current value $\lambda_{i}$ reaches its maximum $\lambda_{\max}$ at such a value of the temperature of the system under study at which the following condition is fulfilled:(8)$$\lambda_{i}=\lambda_{\max}\;\mathrm{at}\;\omega \tau_{i}=1.$$

The frequency ω was determined from the temperature dependencies (Figure 2b and Figure 7b), where
(9)$$\omega =2\pi \nu \Rightarrow \tau_{i_{\max}}=\frac{1}{2\pi \nu_{i_{\max}}},$$
where the relaxation time $\tau_{i_{\max}}$ corresponds to the relaxation time at the peak of dissipative losses in the spectrum and is determined by the Arrhenius equation in the following form [55,65]:(10)$$\tau_{i_{\max}}=\tau_{0}\exp\frac{U_{i_{\max}}}{RT_{i_{\max}}},$$
where $\tau_{0}≃1.6\cdot 10^{-13}s=const$ is the pre-exponential coefficient, $U_{i_{\max}}$ is the activation energy of the relaxation process, and T is the temperature of the system under study.

In this case, the activation energy can be defined as follows:(11)$$U_{i_{\max}}=RT_{i_{\max}}\ln\frac{\tau_{i_{\max}}}{\tau_{0}},$$

Comparative experimental and calculated physical–mechanical and physical–chemical characteristics for the identified processes of dissipative losses in POM are presented in Table 4.

To determine the mechanism of internal friction, for each of the dissipative processes detected on the spectrum of internal friction $\lambda =f\left(T\right)$, the magnitude and sign of the shear modulus defect for this process were calculated from the temperature dependence of the frequency of the oscillatory process (Figure 2 and Figure 7b) according to the following relation:(12)$$\Delta G\left(T\right)=\frac{G_{0}\left(T_{0}\right)-G_{i}\left(T_{i}\right)}{G_{0}\left(T_{0}\right)}=\frac{\nu_{0}^{2}\left(T_{0}\right)-\nu_{i}^{2}\left(T_{i}\right)}{\nu_{0}^{2}\left(T_{0}\right)}.$$

The shear modulus defect can have a positive value for dissipative processes of a relaxation nature and a negative value for dissipative processes of a non-relaxation nature. The calculation of the magnitude and sign of the shear modulus defects for all systems investigated is given in Table 5.

The obtained values of the shear modulus defect make it possible to quantify the real change in the strength characteristics of the materials under study, taking into account local temperature changes in the shear modulus caused by local dissipative losses introduced by each local dissipative process and manifested in the spectrum $\lambda =f\left(T\right)$.

In addition, the values of temperatures, relaxation times (by a ratio of 10), and their ranges at λ=12λmax (Table 6) were determined.

When determining Δτ (*I + II*) of the loss peaks, we found the following:We found the value of the relaxation time of the *I*-peak loss at −92 °C (in calculations, the temperatures were taken in degrees K and the value of the activation energy was taken from Table 4 and was 41.6 kJ/mol);We found the value of *II*-peak loss relaxation time at −48 °C (the value of activation energy was 45.6 kJ/mol);We found the following difference:$$\Delta \tau_{{\left(I+II\right)}_{\max}}=\left(1.6\cdot 10^{-13}\cdot \exp\frac{45.6\cdot 10^{3}}{8.314\cdot (-48+273)}\right)-\left(1.6\cdot 10^{-13}\cdot \exp\frac{41.6\cdot 10^{3}}{8.314\cdot (-92+273)}\right)=0.16\;s.$$The range of Δτ (*IV + V*) of the loss peaks was determined similarly.

The temperature dependence of the shear modulus defect $\Delta G_{i}\left(T\right)$ for each of the detected dissipative processes on the spectrum $\lambda =f\left(T\right)$ allows for a real calculation of the temperature change in the shear modulus $G=f\left(T\right)$ of the polymer system, taking into account the influence of all local anomalous temperature values of the modulus defects $\Delta G_{i}\left(T\right)$, where i≡I, II, III, IV, V, etc.

In addition, the temperature dependence of the shear modulus defect $\Delta G_{i}\left(T\right)$ for each of the detected dissipative processes on the spectrum $\lambda =f\left(T\right)$, together with the given spectra, makes it possible to calculate the relaxation microinheterogenity of each dissipative process detected on the spectrum λiλimax=fT, which has a relaxation mechanism of internal friction.

### Effect of Annealing on Internal Friction Spectra and Temperature Dependencies of Frequency

Let us consider in detail the influence of annealing on the spectra of internal friction and temperature dependencies of the frequency of the free-damping oscillatory process (Figure 8). Both specimens (original and annealed) were cut from the same plate. The experiment on a horizontal torsion pendulum was conducted at a heating rate of 2 °C/min.

In Figure 8 and the values presented in Table 4 and Table 5, we see that annealing does not have a significant effect, namely, the following:There is a slight shift of the peaks along the temperature axis to the right;The intensities of the damping oscillatory process for the corresponding loss peaks remain unchanged;Frequencies at the point of maximum dissipative loss peaks decrease insignificantly;Activation energies and relaxation times increase insignificantly for all selected processes;Values of shear modulus defects for (*I + II*) and (*IV + V*) peaks of dissipative losses are almost unchanged;The temperature range at λ=12λmax varies slightly.

According to the comparative analysis presented above, we see that annealing has no significant effect either visually on the internal friction spectra and temperature–frequency dependencies or on the calculated physical–mechanical and physicochemical characteristics. These judgments are in agreement with [27], where various stabilized POM homopolymers and copolymers were investigated to monitor degradation initiated by processing, artificial aging, and recycling.

However, we notice significant changes in the values of Δ*τ*, (Table 5 and Table 6), which characterize the relaxation microinheterogenity of the system. The (*I + II*) process is characterized by an increase from 0.16 s to 0.40 s, indicating an increase in the number of different subsystems that make up the polymer system. These systems may differ in the values of physical–mechanical and physicochemical characteristics and topology. The (*IV + V*) process is characterized by a not-so-significant increase, namely, from 1.74 s to 2.01 s.

The close values of shear modulus defects Δ*G* of the original and annealed specimens indicate that the changes in elastic characteristics for the original and annealed specimens are insignificant. However, the temperature interval *ΔT* (at λ=12λmax) of the manifestation of the corresponding dissipative process changes significantly, which leads to an increase in the relaxation microheterogeneity of the process. This is confirmed by the values of Δ*τ* in Table 5 and Table 6.

Thus, we conclude that during annealing, the greatest changes take place in the amorphous component of the system, which is apparently due to the appearance of a number of ordered regions in this region (as a result of the lamella thickness growth [66]), which give an increase in the relaxation microinheterogenity (different subsystems making up the new ordered region, subsystems bordering the ordered region and the surrounding amorphous phase, subsystems bordering the region and another ordered region formed next to it, etc.). Consequently, we see that annealing results in an increase in the number of subsystems as well as the formation of subsystems, the defrosting of whose mobility requires more energy costs and longer relaxation time. This is confirmed by the differences in the curves on the temperature–frequency dependencies (Figure 8b).

As mentioned above, changes in the shear modulus defect when comparing the same peaks of unannealed and annealed specimens are insignificant; however, when considering each specimen separately, we see that in the temperature range of manifestation (*I + II*) of the process (in the low-temperature range), the values of the defect in the shear modulus is less than in the region of exhibition temperatures (*IV + V*) (in the region of high temperatures). This suggests that in the low-temperature region, the polymer has a greater ability to resist external influences.

## 5. Conclusions

During the investigation of POM-C by the free-damped oscillatory process, five dissipative loss peaks of different intensities and temperature positions were found, which were characterized by the relaxation nature of internal friction.

The physical–mechanical (shear modulus defect, temperature position of local regions of inelasticity) and physical–chemical (activation energy, discrete relaxation time, intensities of detected dissipative processes) characteristics of each local dissipative process are calculated, and it is concluded that as the temperature increases, the activation energy of the selected dissipative processes increases from 41 to 95 kJ/mol and practically does not depend on annealing; relaxation time also increases from 0.026 s to 0.096 s, but the subsequent heat treatment (annealing) is somewhat more significant.

Annealing has no significant effect on the ability of the material to resist external influences (the maximum variation in the shear modulus defect is 0.06%), spectra and temperature–frequency dependencies (visually), and physical–mechanical characteristics. It has been found that the intensities of dissipative processes remain virtually unchanged for both annealed and non-annealed samples. Additionally, according to computational data, small changes are also characteristic of the following parameters: the activation energy varies from 0.5 to 1.4 kJ/mol and the relaxation time changes from 0.002 to 0.007 s, depending on the presence or absence of annealing. As a result of annealing, there is a significant increase in the relaxation microinheterogenity of the polymer system across the entire temperature range. The greatest changes are characteristic for the low-temperature region, 250%; for the high-temperature region, this indicator is 115%.

## Figures and Tables

**Figure 1 polymers-16-03582-f001:**
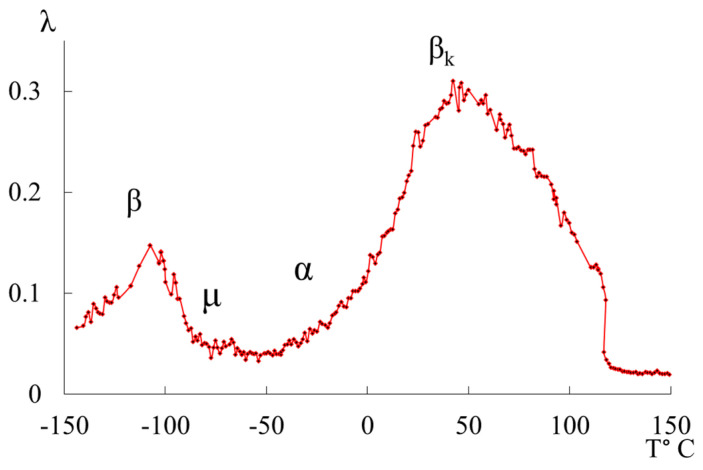
Internal friction spectrum $\lambda =f\left(T\right)$ of HDPE.

**Figure 2 polymers-16-03582-f002:**
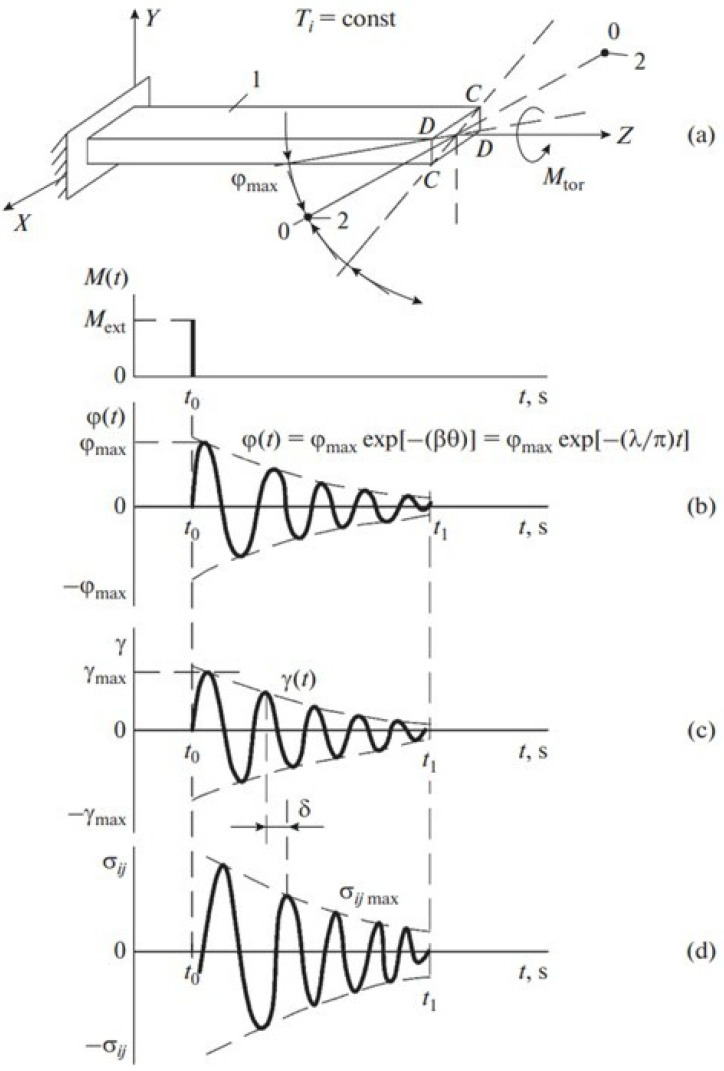
A schematic diagram describing in general terms a free-damping oscillatory process excited in the material under study; (**a**) in isothermal mode T=const by pulse action. Sweep of the time dependence of the twist angle $\varphi \left(t\right)$ relative to the longitudinal axis Z of the specimen—(**b**). The deformation of the sample—$\gamma \left(t\right)$ (**c**) and the corresponding shear stresses $\sigma_{ij}$ occurring in the sample—(**d**). β—damping coefficient of the oscillatory process; θ— period of the vibration process. All other designations are defined below in the text of the article [53,54].

**Figure 3 polymers-16-03582-f003:**
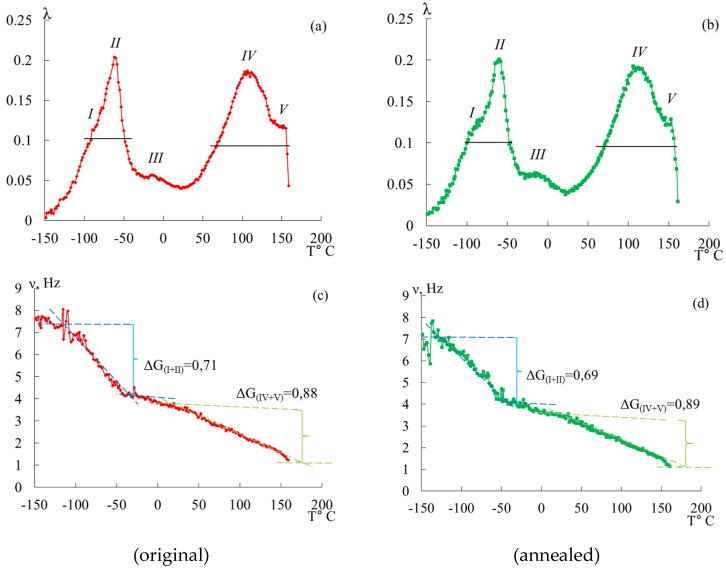
Internal friction spectra $\lambda =f\left(T\right)$ and temperature dependency of frequency $\nu =f\left(T\right)$ of the free-damped oscillatory process for original (**a,c**) and annealed (**b**,**d**) POM specimens.

**Figure 4 polymers-16-03582-f004:**
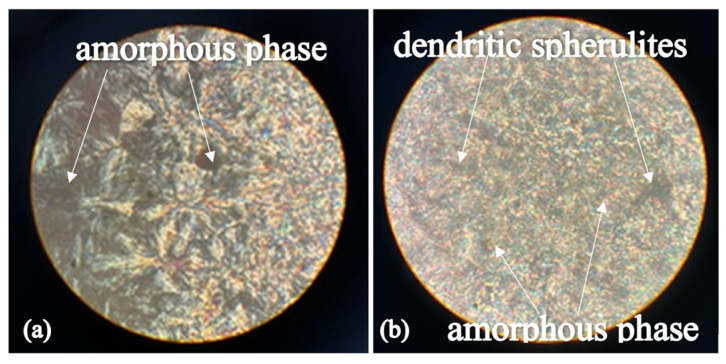
Structure of specimens of amorphous–crystalline polyoxymethylene films: (**a**) Maltese cross; (**b**) dendritic spherulites. Resolution: 70 × 16.

**Figure 5 polymers-16-03582-f005:**
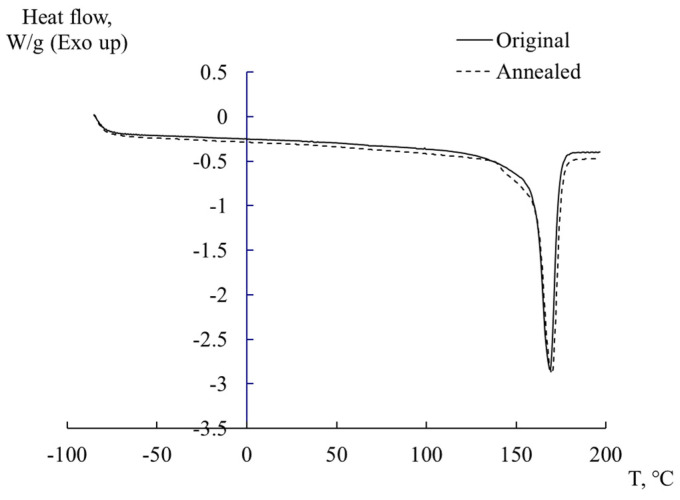
DSC curves for both the original and annealed POM specimens.

**Figure 6 polymers-16-03582-f006:**
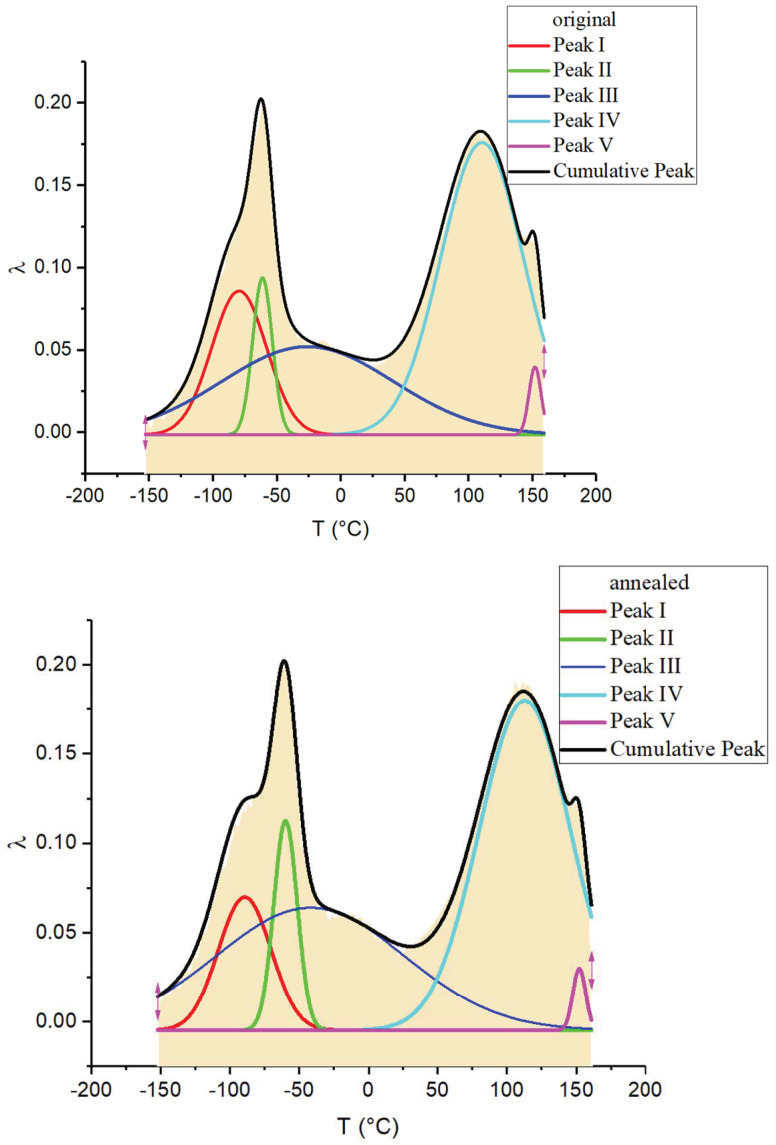
Expansion of dissipative loss peaks using the Gaussian normal distribution for the original and annealed specimens.

**Figure 7 polymers-16-03582-f007:**
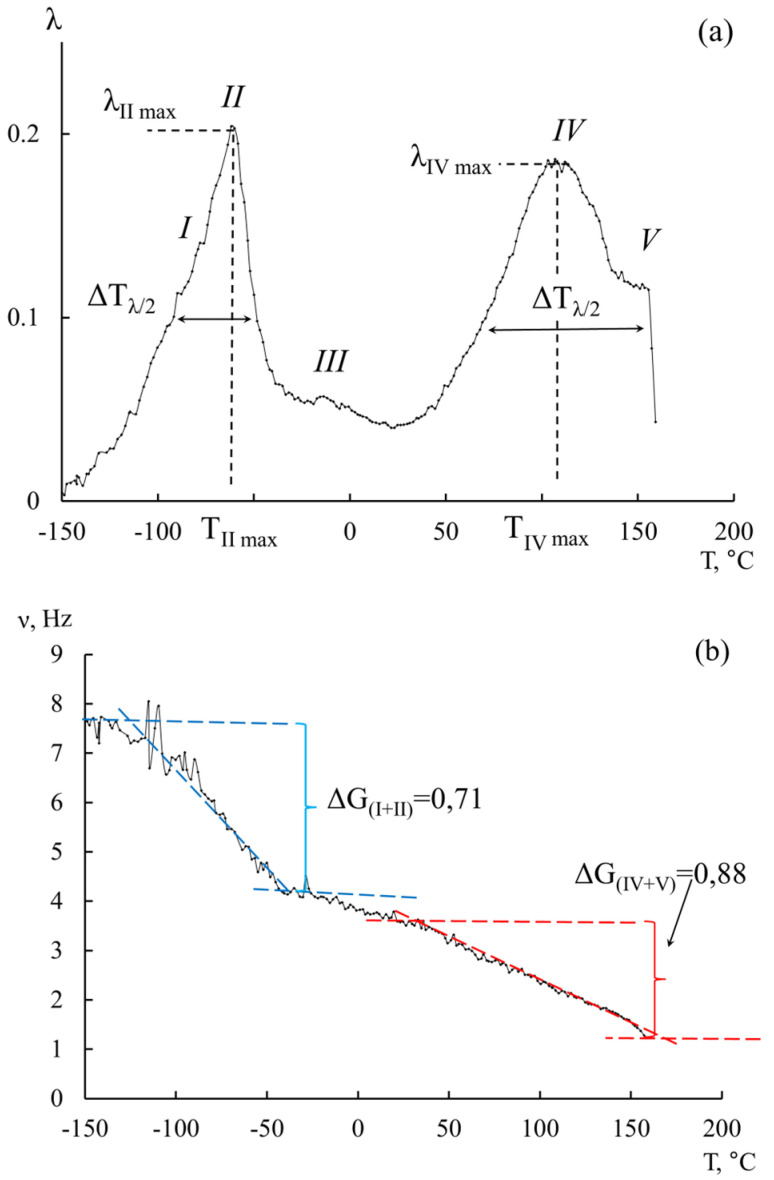
Internal friction spectrum (**a**) and temperature–frequency dependence (**b**) for the original POM-27 specimen.

**Figure 8 polymers-16-03582-f008:**
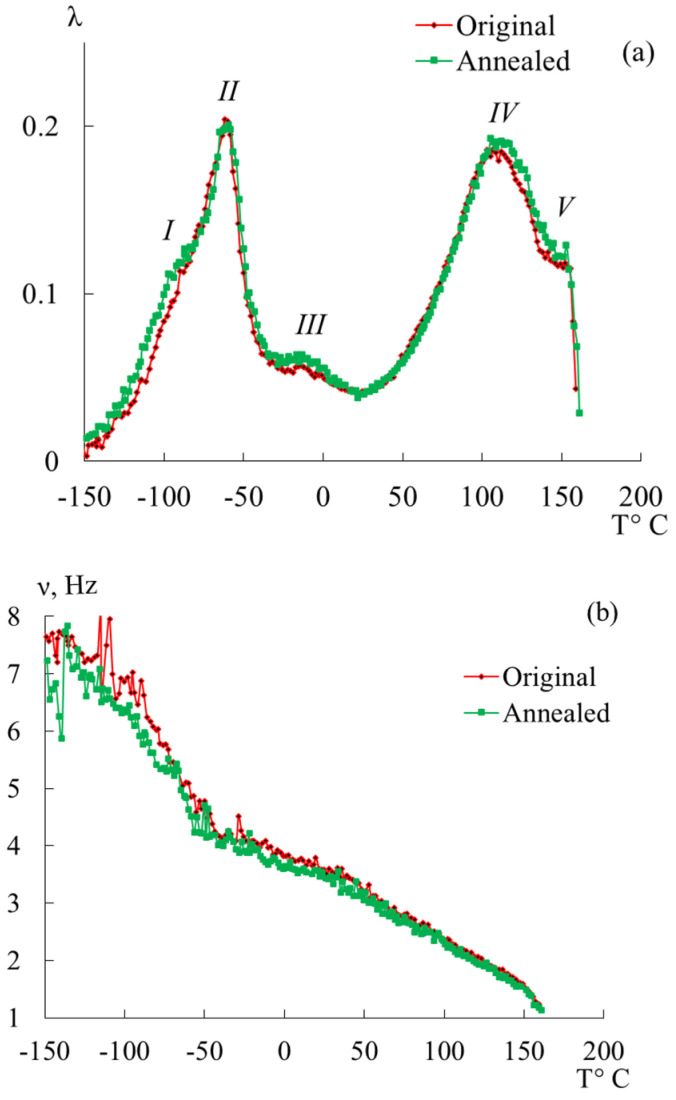
Generalized spectrum of internal friction (**a**) and temperature–frequency dependence (**b**) for POM-27 original and annealed specimens.

**Table 1 polymers-16-03582-t001:** Comparative characteristics of POM-H and POM-C [29,30].

Characteristic	POM-H	POM-C
Melting point, °C	172–184	160–175
Processing temperature, °C	194–244	172–205
Elastic modulus, MPa	4623	3105
Tensile strength, MPa	70	61
Elongation at break, %	25	40–75
Glass transition temperature (Tg), °C	−85	−60
Density, g/cm^3^	1.41	1.34

**Table 2 polymers-16-03582-t002:** Main characteristics of Mascon POM 27 [48].

Properties	Values	Unit of Measure
Physical properties		
Density	1410	kg/cm^3^
Melt flow index	27	g/10 min
Volumetric melt flow index	25	cm^3^/10 min
Mechanical properties		
Tensile modulus (1 mm/min)	2700	MPa
Tensile stress, 50 mm/min	65	MPa
Flexural modulus, 23 °C	2550	MPa
Charpy V-notch impact energy, 23 °C	5	kJ/m^2^
Melting point *, 20 °C/min	169	°C
Thermal deformation temperature, 1.8 MPa	95	°C
Degree of crystallinity *	53	%
Average molecular weight, M_w_ *	6.5 × 10^4^	-

* Determined from the experimental results of this paper.

**Table 3 polymers-16-03582-t003:** Experimental values of temperature and enthalpy of fusion and calculated values of the degree of crystallinity for the original and annealed POM.

Specimen	T_m_, °C	ΔH_f_, J/g	χ, %
Original	169	173.91	53.35
Annealed	170	167.59	51.41

**Table 4 polymers-16-03582-t004:** Basic physical–mechanical and physical–chemical characteristics for all dissipative loss processes.

**Specimen**	T_max_ (K)	T_max_ (°C)	$\lambda_{\max}$	$\begin{array}{l} \nu_{\max},\\ \mathrm{Hz} \end{array}$	$U_{i_{\max}}$, kJ/mol	$\tau_{i_{\max}},s$
Peak *I*						
original	193.9	−79	0.14	6.04	41.6	0.026
annealed	190.9	−82	0.13	5.62	41.1	0.028
Peak *II*						
original	211.2	−62	0.20	5.11	45.6	0.031
annealed	213.2	−60	0.20	4.63	46.2	0.034
Peak *III*						
original	295.2	−22	0.05	4.22	64.3	0.038
annealed	295.0	−21	0.06	3.88	64.4	0.041
Peak *IV*						
original	379.8	107	0.19	2.25	84.7	0.071
annealed	385.0	112	0.19	2.08	86.1	0.077
Peak *V*						
original	417.8	145	0.12	1.66	94.2	0.096
annealed	421.0	148	0.12	1.55	95.2	0.103

**Table 5 polymers-16-03582-t005:** Experimental values of temperatures and frequencies and calculated values of modulus defects of the original and annealed specimens.

Specimen	T_max_ (°C)		The Range of Frequency Variation, Hz.		Shear Modulus Defect Δ*G*	ΔT(°C)	Δτ,s
						at λ = 12λmax	
Peak (*I + II*)							
original	−123	−39	7.65	4.14	0.71	44	0.16
annealed	−133	−38	7.23	4.00	0.69	52	0.40
Peak (*IV + V*)							
original	30	168	3.54	1.23	0.88	94	1.78
annealed	27	172	3.42	1.14	0.89	86	2.06

**Table 6 polymers-16-03582-t006:** Experimental values of temperatures, relaxation times, and their ranges at λ=12λmax for original and annealed samples.

Specimen	T, (°C)		τ,s		ΔT, (°C)	Δτ,s
Peak (*I + II*)						
original	−123	−39	7.65	4.14	44	0.16
annealed	−133	−38	7.23	4.00	52	0.40
Peak (*IV + V*)						
original	30	168	3.54	1.23	94	1.78
annealed	27	172	3.42	1.14	86	2.06

## Data Availability

Data are contained within the article.

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
