# Peer review of "Temperature Areas of Local Inelasticity in Polyoxymethylene"

_polymers, 2024, doi:10.3390/polym16243582_

Round 1
Reviewer 1 Report
Comments and Suggestions for Authors
1. Quantify results in abstract and conclusion
2. The authors had given references such as “…. polyethylene (PE) [1–7]. Both materials have their end-use advantages and disadvantages. [1,5,6,8–12]. Don’t cluster the references.
3. What is the literature gap identified?
4. State the novelty of the present study, and how it differs from the available literature?
5. Improve the quality of the presented figures.
The detailed comments are:1. Quantify results in abstract and conclusion
2. The authors had given references such as “…. polyethylene (PE) [1–7]. Both materials have their end-use advantages and disadvantages. [1,5,6,8–12]. Don’t cluster the references.
3. Literature section needs to be augmented with research related to this study.
4. Table 1 and associated explanations can be moved to materials section 2.
5. In Table 1 check the unit for density. In cm3, 3 must be in superscript.
6. Introduction section is written as a thesis. It must be rewritten properly for the intended purpose.
7. What is the literature gap identified?
8. State the novelty of the present study, and how it differs from the available literature?
9. In section 2.1, it is given that, “the object of the study”. It must be changed to material.
10. Check for typographical errors and grammatical errors. For example, “where is n the number of oscillations….”.
11. In section 3, it is given that, “in this paper, the following experiments were carried out”. This must be avoided in research articles.
12. Provide proper references to the equations presented in the manuscript.
13. In Figure 3, superimpose the two material outcomes for better comparison.
14. Shorten the section heads. For example, “section 3.2… is very lengthy”
15. Annotate OMs in Figure 4.
16. Figure 5 does not make any variation between the original and annealed samples. Why?
17. In Figure 6, superimpose all the relevant data’s between original and annealed samples for better presentation.
18. In Table 5, for both samples, very near values are obtained. Justify
19. Improve the quality of the presented figures.
20. Conclusion must be more concise with the findings. Quantify results in conclusion.
Author Response
We thank the reviewer for their time and helpful comments - the feedback has greatly improved this article. Our responses to all comments are on file (reviewer comments are in bold). All changes to the text of the article are color-coded: green is what has been added; yellow is what is to be deleted.
Please see the attachment.

Reviewer 2 Report
Comments and Suggestions for Authors
1. In Introduction, the authors point out the problems that have existed in previous studies, so have these problems been studied by other researchers? And then how did they study it? How much did they study it? This content should focus on the description.
Does the author's research task revolve around the above questions?
2. In section 3, the title length of sub-section is too long.
3. The experimental method for each experiment, such as amorphous-crystalline structure, DSC, should be described in section 2.
4. The separation rate of all figures needs to be improved.
5. In section 4, author should cite relevant literature, and carry out comparative discussion and analysis.
6. The references are too old, at least half of the references should be published in the last five years.
Author Response

(The authors gave the same response as above.)

Round 2
Reviewer 1 Report
Comments and Suggestions for Authors
1. Quantify results in abstract and conclusion sections, such that how much variation (in %) exists between the samples considered for each test, which will improve the presentation.
2. Don’t cluster the references [1-7], and [8-12]. Cluster a maximum of [1-3] references.
3. Annotate inside the OM presented in Figure 4 showing the amorphous phases and dendritic spherulites formed.
Author Response
We thank the reviewers for their time and helpful comments - their feedback has greatly improved this article. Below are our responses to each comment (reviewers' comments are in bold). All changes to the text of the article are color-coded (green).
1. Quantify results in abstract and conclusion sections, such that how much variation (in %) exists between the samples considered for each test, which will improve the presentation.
Response: The authors took into account the reviewer's comments and made corrections to the manuscript.
The changes in activation energies and relaxation times are given not in relative values (in %) but in absolute values (kJ/mol for activation energies and seconds for relaxation times) because the changes in absolute values are small. However, due to the difference in activation energy values being approximately 2 times and relaxation times being approximately 4 times for processes I and IV, relative values may be misleading. All values and calculations are presented in detail in the tables.
2. Don’t cluster the references [1-7], and [8-12]. Cluster a maximum of [1-3] references.
Response: The authors took into account the reviewer's comments and made corrections to the manuscript.
3. Annotate inside the OM presented in Figure 4 showing the amorphous phases and dendritic spherulites formed.
Response: The authors took into account the reviewer's comments and made corrections to the manuscript.
Reviewer 2 Report
Comments and Suggestions for Authors
The author has made careful revisions and the revised version is acceptable for publication.
Author Response
The authors would like to thank the reviewer for the careful reading of the work and for the comments provided.